# Are Adaptive Chemotherapy Schedules Robust? A Three-Strategy Stochastic Evolutionary Game Theory Model

**DOI:** 10.3390/cancers13122880

**Published:** 2021-06-09

**Authors:** Rajvir Dua, Yongqian Ma, Paul K. Newton

**Affiliations:** 1Department of Mathematics, University of Southern California, Los Angeles, CA 90089-1191, USA; rajvirdu@usc.edu; 2Department of Physics & Astronomy, University of Southern California, Los Angeles, CA 90089-1191, USA; yongqiam@usc.edu; 3Department of Aerospace & Mechanical Engineering, Mathematics, The Ellison Institute, University of Southern California, Los Angeles, CA 90089-1191, USA

**Keywords:** chemotherapy schedules, adaptive chemotherapy, uncertainty quantification, Moran process model, evolutionary game theory model, evolutionary cycles, tumor chemoresistance

## Abstract

**Simple Summary:**

We describe an evolutionary game theory mathematical model to investigate the robustness and accuracy of adaptive chemotherapy schedules in a stochastic environment in a tumor. The model assumes tumors are made up of a finite cell population of chemo-sensitive cells, and two populations of resistant cells, each resistant to one of two separate drugs that can be administered on different schedules. The goal of the adaptive schedules are to delay chemoresistance in the tumor by keeping the cell populations in competition with each other without any of the populations reaching fixation leading to treatment failure.

**Abstract:**

We investigate the robustness of adaptive chemotherapy schedules over repeated cycles and a wide range of tumor sizes. Using a non-stationary stochastic three-component fitness-dependent Moran process model (to track frequencies), we quantify the variance of the response to treatment associated with multidrug adaptive schedules that are designed to mitigate chemotherapeutic resistance in an idealized (well-mixed) setting. The finite cell (*N* tumor cells) stochastic process consists of populations of chemosensitive cells, chemoresistant cells to drug 1, and chemoresistant cells to drug 2, and the drug interactions can be synergistic, additive, or antagonistic. Tumor growth rates in this model are proportional to the average fitness of the tumor as measured by the three populations of cancer cells compared to a background microenvironment average value. An adaptive chemoschedule is determined by using the N→∞ limit of the finite-cell process (i.e., the adjusted replicator equations) which is constructed by finding closed treatment response loops (which we call evolutionary cycles) in the three component phase-space. The schedules that give rise to these cycles are designed to manage chemoresistance by avoiding competitive release of the resistant cell populations. To address the question of how these cycles perform in practice over large patient populations with tumors across a range of sizes, we consider the variances associated with the approximate stochastic cycles for finite *N*, repeating the idealized adaptive schedule over multiple periods. For finite cell populations, the distributions remain approximately multi-Gaussian in the principal component coordinates through the first three cycles, with variances increasing exponentially with each cycle. As the number of cycles increases, the multi-Gaussian nature of the distribution breaks down due to the fact that one of the three sub-populations typically saturates the tumor (competitive release) resulting in treatment failure. This suggests that to design an effective and repeatable adaptive chemoschedule in practice will require a highly accurate tumor model and accurate measurements of the sub-population frequencies or the errors will quickly (exponentially) degrade its effectiveness, particularly when the drug interactions are synergistic. Possible ways to extend the efficacy of the stochastic cycles in light of the computational simulations are discussed.

## 1. Introduction

The design of adaptive chemotherapy schedules [1], motivated and aided by mathematical/computational models [2], is a rapidly developing field that has great potential for mitigating chemoresistance in tumors [3]. The advantage of an adaptive schedule over a more widely accepted pre-determined schedule, such as a maximum tolerated dose (MTD) schedule, or a low-dose metronomic (LDM) schedule [4,5], is that adaptive schedules are able to evolve and change along with the tumor [6]. However, as has been observed [7], the efficacy of sequential adaptive cycles depends crucially on the ability to monitor and track the balance of heterogeneous sub-populations of cells that comprise the tumor [8], or at least an accurate surrogate biomarker [9,10], which can be challenging. Additionally, the fact that any complex finite population of cells will evolve stochastically [11,12,13] makes it particularly important to assess the effectiveness of adaptive schedules under a range of diverse conditions, such as different patient populations and tumor sizes and over many therapy cycles. We present a mathematical model that addresses these issues.

Standard pre-scheduled chemotherapy dose-delivery schedules suffer from the common occurrence of chemoresistance of the tumor [14], leading to treatment failure [15] after multiple cycles. To overcome this, adaptive schedules are designed with the goal of managing resistance [1]. Adaptive therapies typically leverage tumor heterogeneity [8] by exploiting evolution via competition of the tumor sub-populations [16] to steer evolution and designing an advantageous chemoschedule [17,18,19] that delays chemoresistance by maintaining a sufficient fraction of the sensitive population in order to suppress the resistant population, a strategy used in the context of avoiding antibiotic resistance [20] as well.

Aside from schedule design (dose and timing), the use of two or more drugs is helpful in mitigating resistance, if combined strategically [2,21]. A key challenge associated with dose scheduling and the optimal design of multidrug combinations is that both the drug schedules and the multidrug mixture rely on an accurate dynamical assessment of the relative balance and mixture of the evolving cell types in the tumor, which is hard to obtain in clinical practice [7]. Quantitative assessments are, generally speaking, much easier to obtain in a mathematical model.

Even within the framework of a mathematical model, how exactly to quantify the effectiveness of the schedule and multidrug mixture when the schedule is adaptive (i.e., not fixed and repeatable) is not at all straightforward. We address this important issue of uncertainty quantification and robustness [22,23,24] of tumor response to adaptive therapy schedules and synergystic vs. antagonistic multidrug interactions [25,26] by using a stochastic finite-cell fitness-dependent Moran process evolutionary game theory tumor model with an adaptive schedule designed from the deterministic adjusted replicator dynamical system [27], which is the large cell limit (N→∞) of the finite-cell stochastic process [28,29]. We describe the main features of our model as well as the connections between the finite cell stochastic model and infinite cell deterministic model in the next section.

## 2. Model Description

The model we use to investigate the potential efficacy of designing closed evolutionary cycles to avoid chemoresistance in a stochastic setting consists of the following basic elements:1.We implement a version of the adjusted replicator dynamics equations to track the frequency evolution of the three sub-populations of cancer cells that make up the tumor. Our model consists of a chemo-sensitive population (S), a population resistant to drug 1 ($R_{1}$), and a population resistant to drug 2 ($R_{2}$). The competition among the three populations is determined by a 3×3 payoff matrix *A* that builds in a fitness cost of resistance. In the absence of chemotherapy, the sensitive population is most fit and will outcompete the resistant populations, saturating the tumor.2.We use two time-dependent control functions $C_{1}\left(t\right)$ and $C_{2}\left(t\right)$ to model the chemotherapy dosing schedules. These chemotherapy dosing functions control three selection pressure parameters $w_{1}\left(t\right)$, $w_{2}\left(t\right)$, and $w_{3}\left(t\right)$ by altering the relative fitness values of the three sub-populations. A dose of chemotherapy ($C_{i}\left(t\right)>0;t_{0}\leq t\leq t_{1}$) lowers the relative fitness of the targeted cell population by altering the selection pressures on the sub-populations to effectively favor ones that are not targeted. This mechanism allows us to ‘design’ favorable fitness landscapes indirectly by adaptively monitoring the sub-population frequencies and altering our dosing schedule in response. Our goals when we design chemotherapy schedules are to: (i) avoid fixation of the sensitive cell population; (ii) avoid chemoresistance (fixation of either of the resistant populations) by keeping the three populations of cells in competition without allowing any of them to saturate the tumor. We implement this by designing schedules that keep us confined to a closed ‘evolutionary cycle’ which keep the sub-populations in competition forever (ideally) for the deterministic (N=∞) model.3.We then test the performance of these designed schedules on a finite cell (N<∞) stochastic Moran process model to track the sub-population frequencies during these cycles, where the limit N→∞ corresponds to our adjusted replicator model from which the cycle was designed.4.Since the Moran process model uses a fixed value of *N* (hence, it cannot be used directly to determine tumor growth where the total cancer cell population increases), we add a tumor growth equation with growth rate determined as a function of the average fitness of the three cancer cell sub-population frequencies comprising the tumor. When the average fitness of the cancer cell populations is above a fixed microenvironmental average, the tumor grows (exponentially), and when it is below, it shrinks (exponentially).

Details of each of these elements are described in the next section.

### 2.1. Three-Component Two-Drug Adjusted Replicator Dynamics Model

The deterministic model we use to determine the sub-population frequencies $\left(x_{1},x_{2},x_{3}\right)\equiv \left(S,R_{1},R_{2}\right)$ and the dosing schedules $\left(C_{1}\left(t\right),C_{2}\left(t\right)\right)$ to determine the adaptive therapy are the adjusted replicator equations:(1)$$\begin{array}{ccc} {\dot{x}}_{\alpha } & = & \frac{\left(f_{\alpha }-\langle f\rangle \right)}{\langle f\rangle }x_{\alpha }\;\left(\alpha =1,2,3\right) \end{array}$$(2)$$\begin{array}{ccc} x_{1} & + & x_{2}+x_{3}=1 \end{array}$$

Here, $f_{\alpha }$ denotes the fitness of sub-population α:(3)$$\begin{array}{ccc} f_{\alpha } & = & 1-w_{\alpha }+w_{\alpha }{\left(A\vec{x}\right)}_{\alpha }\;\left(\alpha =1,2,3\right) \end{array}$$
while 〈f〉 denotes the average fitness of the entire population (tumor):(4)$$\begin{array}{ccc} \langle f\rangle & = & f_{1}x_{1}+f_{2}x_{2}+f_{3}x_{3} \end{array}$$

We restrict $0\leq w_{\alpha }\leq 1$ and note that for $w_{\alpha }=0$, fitness is constant ($f_{\alpha }=1$), while for $w_{\alpha }=1$, fitness is determined by the payoff matrix *A*. The growth or decay of population α in Equation (Equation 1) is determined by the deviation of the fitness of population α from the average, normalized by the average fitness of the tumor.

The relative fitness values of the three sub-populations are controlled by the selection pressure parameters $w_{\alpha }$: (5)$$\begin{array}{ccc} w_{1}\left(t\right) & = & w_{0}\left(1-C_{1}\left(t\right)-C_{2}\left(t\right)-eC_{1}\left(t\right)C_{2}\left(t\right)\right) \end{array}$$(6)$$\begin{array}{ccc} w_{2}\left(t\right) & = & w_{0}\left(1-C_{2}\left(t\right)\right) \end{array}$$(7)$$\begin{array}{ccc} w_{3}\left(t\right) & = & w_{0}\left(1-C_{1}\left(t\right)\right), \end{array}$$
where $\vec{C}\left(t\right)=\left(C_{1}\left(t\right),C_{2}\left(t\right)\right)$ is the chemotherapy delivery function, with $C_{1}$ controlling drug 1, $C_{2}$ controlling drug 2, and *e* is a parameter that determines whether the two drugs act synergistically (e>0), antagonistically (e<0), or additively (e=0). We take $w_{0}=0.1$, which sets the timescale in our simulations. Notice that when drug 1 is applied ($C_{1}\left(t\right)>0$), it acts to reduce the selection pressure parameters $w_{1},w_{3}$, but leaves $w_{2}$ (the parameter controlling the $R_{1}$ population) unchanged, so the $R_{1}$ population is resistant to drug 1. When drug 2 is applied ($C_{2}\left(t\right)>0$), it acts to reduce the selection pressure parameters $w_{1},w_{2}$, leaving $w_{3}$ unchanged (the parameter controlling the $R_{2}$ population), so the $R_{2}$ population is resistant to drug 2. Thus, our model steers evolution by exploiting the effect that chemotherapy has on the selection pressure applied to different sub-populations. The total dose delivered over time-period *t* is denoted $\vec{D}\in R^{2}$: (8)$$\begin{array}{ccc} \vec{D}\left(t\right) & = & \left(D_{1}\left(t\right),D_{2}\left(t\right)\right)=\int_{0}^{t}\vec{C}\left(t\right)dt,\;\vec{D}\left(0\right)=0. \end{array}$$

Then: (9)$$\begin{array}{c} \dot{\vec{D}}\left(t\right)=\vec{C}\left(t\right) \end{array}$$
and: (10)$$\begin{array}{ccc} \vec{D}\left(T\right) & = & \int_{0}^{T}\vec{C}\left(t\right)dt={\vec{D}}_{T} \end{array}$$

${\vec{D}}_{T}$ denotes the total dose delivered in fixed time *T*. The system (Equation 1) is a nonlinear non-constant coefficient dynamical system governing the frequency distributions.

To fix the evolutionary game being played by the population of cells, we consider the 3×3 payoff matrix *A*:(11)$$\begin{array}{ccc} A & = & \left[\begin{array}{ccc} a_{11} & a_{12} & a_{13}\\ a_{21} & a_{22} & a_{23}\\ a_{31} & a_{32} & a_{33} \end{array}\right] \end{array}$$

In order to model tumor kinetics, we take our payoff matrix to be of Prisoner’s Dilemma (PD) type [30]. The reasons why the PD matrix is useful as a cancer model are discussed in more detail in [19,30], but arise from the fact that this choice allows the sensitive cell population (the defectors) to reach fixation with an *S*-shaped growth curve in the absence of chemotherapy $(C_{1}\left(t\right),C_{2}\left(t\right))=\left(0,0\right)$ and enforces a ’cost-of-resistance’ on the two resistant sub-populations (the cooperators) [31]. The PD inequality bounds for the entries of the payoff matrix are:(12)$$\begin{array}{c} a_{21}<a_{11}<a_{22}<a_{12} \end{array}$$(13)$$\begin{array}{c} a_{31}<a_{11}<a_{33}<a_{13} \end{array}$$(14)$$\begin{array}{c} a_{32}<a_{22}<a_{33}<a_{23} \end{array}$$

For definiteness, we choose the specific values:(15)$$A=\left[\begin{array}{ccc} 2 & 2.8 & 2.8\\ 1.5 & 2.1 & 2.3\\ 1.5 & 1.8 & 2.2 \end{array}\right]$$

Our chemotherapy delivery functions are constrained so that $0\leq C_{1}\left(t\right)\leq 1$, $0\leq C_{2}\left(t\right)\leq 1$, $0\leq C_{1}\left(t\right)+C_{2}\left(t\right)\leq 1$, and from Equations (Equation 5)–(7) reduce the fitness of the relevant sub-populations by altering the relative selection pressures.

Since the Moran process model tracks frequencies while keeping *N* fixed, we need an additional tumor growth equation to complete our model. For this, we use the following equation to track the tumor volume V(t):(16)$$\dot{V}=\left(\langle f\rangle -g\right)V$$

Here, *g* denotes a positive constant we take as generally representing the fitness of the average microenvironment surrounding the tumor. Hence, the growth rate (or decay) of the tumor, (〈f〉−g), is given by the deviation of the average tumor fitness from the average fitness of the local microenvironment, as represented by the gray region surrounding the tumor cells in Figure 1.

### 2.2. Three-Component Two-Drug Discrete Stochastic Moran Process

In more detail, our finite cell model is a three component stochastic birth–death process [32,33], with frequency-dependent fitness governed by a payoff matrix, and population size *N* comprised of a fluctuating mixture of cells of type *S* (sensitive), cells of type $R_{1}$ (resistant to drug 1), and cells of type $R_{2}$ (resistant to drug 2), with $S+R_{1}+R_{2}=N$. At every step *n* in the process, a cell is randomly selected for reproduction with probability proportional to its fitness. The offspring produced replaces a randomly chosen cell in the population. The fitness of cells in the sub-populations *S*, $R_{1}$, and $R_{2}$ are given respectively as:(17)$$\begin{array}{ccc} f_{ij} & = & 1-w_{1}\\ & + & w_{1}\left[a_{11}\frac{i-1}{N-1}+a_{12}\frac{j}{N-1}+a_{13}\frac{N-i-j}{N-1}\right] \end{array}$$(18)$$\begin{array}{ccc} g_{ij} & = & 1-w_{2}\\ & + & w_{2}\left[a_{21}\frac{i}{N-1}i+a_{22}\frac{j-1}{N-1}+a_{23}\frac{N-i-j}{N-1}\right] \end{array}$$(19)$$\begin{array}{ccc} h_{ij} & = & 1-w_{3}\\ & + & w_{3}\left[a_{31}\frac{i}{N-1}+a_{32}\frac{j}{N-1}+a_{33}\frac{N-i-j-1}{N-1}\right]. \end{array}$$

Here, *i* denotes the number of cells of type *S*, *j* denotes the number of cells of type $R_{1}$, and N−i−j denotes the number of cells of type $R_{2}$, with 0≤i≤N;0≤j≤N,i+j≤N. The state of the system is given by (i,j) (visualized as a grid point on a triangular grid as in Figure 1) with probability of residing in that state after *n* steps given by $p_{ij}^{\left(n\right)}$.

As in the continuous model, the parameter $w_{\alpha }\in \left[0,1\right]$ (α=1,2,3) controls the selection pressure and allows us to independently adjust the fitness landscape. At the two extremes, when $w_{\alpha }=0$, the payoff matrix makes no contribution to fitness, so only random drift governs the dynamics. At the other extreme, when $w_{\alpha }=1$, the payoff matrix makes a large contribution to the fitness, with selection pressure driving the dynamics. We define the average tumor fitness, ${\langle f\rangle }_{N}$, as the discrete analogue of (Equation 4):(20)$$\begin{array}{ccc} {\langle f\rangle }_{N} & = & \frac{S_{ij}}{N} \end{array}$$
where:(21)$$\begin{array}{c} S_{ij}=if_{ij}+jg_{ij}+\left(N-i-j\right)h_{ij} \end{array}$$

The transition probabilities at each step of the birth–death (Markov) process are written as: (22)$$\begin{array}{ccc} p_{ij}^{SR_{1}} & = & \frac{if_{ij}}{S_{ij}}\frac{j}{N} \end{array}$$(23)$$\begin{array}{ccc} p_{ij}^{R_{1}S} & = & \frac{jg_{ij}}{S_{ij}}\frac{i}{N} \end{array}$$(24)$$\begin{array}{ccc} p_{ij}^{SR_{2}} & = & \frac{if_{ij}}{S_{ij}}\frac{N-i-j}{N} \end{array}$$(25)$$\begin{array}{ccc} p_{ij}^{R_{2}S} & = & \frac{\left(N-i-j\right)h_{ij}}{S_{ij}}\frac{i}{N} \end{array}$$(26)$$\begin{array}{ccc} p_{ij}^{R_{1}R_{2}} & = & \frac{jg_{ij}}{S_{ij}}\frac{N-i-j}{N} \end{array}$$(27)$$\begin{array}{ccc} p_{ij}^{R_{2}R_{1}} & = & \frac{\left(N-i-j\right)h_{ij}}{S_{ij}}\frac{j}{N} \end{array}$$(28)$$\begin{array}{ccc} p_{ij}^{const} & = & p_{ij}^{SS}+p_{ij}^{R_{1}R_{1}}+p_{ij}^{R_{2}R_{2}}\\ & = & 1-(p_{ij}^{SR_{1}}+p_{ij}^{R_{1}S}+p_{ij}^{SR_{2}}+p_{ij}^{R_{2}S}+p_{ij}^{R_{1}R_{2}}+p_{ij}^{R_{2}R_{1}}) \end{array}$$

To implement chemotherapy in the discrete Markov process, we discretize time 0≤t≤T in (8)–(10) so that at each time-step *n* in the process (with n=tN), we enforce the time-dependent chemotherapy schedules (Equation 5)–(10) by adjusting the selection pressures where: (29)$$\begin{array}{ccc} w_{1}^{n} & = & w_{0}\left(1-C_{1}^{n}-C_{2}^{n}-eC_{1}^{n}C_{2}^{n}\right) \end{array}$$(30)$$\begin{array}{ccc} w_{2}^{n} & = & w_{0}\left(1-C_{1}^{n}\right) \end{array}$$(31)$$\begin{array}{ccc} w_{3}^{n} & = & w_{0}\left(1-C_{2}^{n}\right), \end{array}$$
(32)$$\begin{array}{ccc} {\vec{D}}^{n} & = & \left(D_{1}^{n},D_{2}^{n}\right) \end{array}$$
(33)$$\begin{array}{ccc} & = & \sum_{k=0}^{n-1}\left(C_{1}^{k},C_{2}^{k}\right). \end{array}$$

This renders the Markov process non-stationary [34]. The state-vector ${\vec{P}}^{n}\in R^{(N+2)(N+1)/2}$ represents the discrete probability distribution at each triangular grid point (i,j), $p_{ij}^{\left(n\right)}$ (the finite analogue of the Master equation and Fokker-Planck formulation in [29]):(34)$$\begin{array}{ccc} {\vec{P}}^{n} & = & {\left[p_{00}^{\left(n\right)},p_{01}^{\left(n\right)},...,p_{0N}^{\left(n\right)},...,p_{N0}^{\left(n\right)}\right]}^{T} \end{array}$$

Along with a $\frac{1}{2}\left(N+2\right)\left(N+1\right)\times \frac{1}{2}\left(N+2\right)\left(N+1\right)$ stochastic (non-stationary) transition matrix $M\equiv \{p_{ij}\}$ (whose entries are Equations (Equation 22)–(31)) with rows adding to 1 driving the dynamical system:(35)$${\vec{P}}^{n+1}=M{\vec{P}}^{n}.$$

We initiate the process with an initial discrete distribution ${\vec{P}}^{0}$. This framework allows us to carry out Monte Carlo simulations to generate probability distributions (using Equations (Equation 22)–(27) more practically) of the three cell types for any initial distribution.

### 2.3. Continuous Limit N→∞ Which Relates the Moran Process to the Adjusted Replicator System

The deterministic and stochastic models are connected by the limit N→∞ in which the discrete finite cell frequency-dependent model (Equation 35) converges to the adjusted replicator system (Equation 1). This connection was first made explicit in [28,29] with an earlier approach in [35]. These papers accomplished two things. First, the authors of [28] showed that different microscopic stochastic processes for two-strategy games lead to either the replicator system or the adjusted replicator system in the limit N→∞. They also derived an explicit mean-field Fokker–Planck equation describing the probability density function for the different sub-populations. This work was extended in [29] to an arbitrary number of strategies (sub-populations), with the possible inclusion of mutations leading to the adjusted replicator dynamics as N→∞. Our method of quantifying the uncertainty associated with adaptive chemotherapy schedules exploits this stochastic–deterministic connection. The deterministic adjusted replicator Equation (Equation 1) allows us to unambiguously design a beneficial schedule giving rise to the closed evolutionary cycle, while the stochastic fitness-dependent three-strategy Moran process allows us to quantify the variance associate with repeated cycling of the schedule for finite *N*.

## 3. Results

To understand our results, we start with an example in Figure 2a of one (of many) closed deterministic evolutionary cycle ABCA in the $\left(S,R_{1},R_{2}\right)$ tri-linear plane, along with two stochastic realizations starting at the same point *A* but not closing the loop. The deterministic cycle is exactly closed and repeatable, but the probability of a stochastic cycle being closed is vanishingly small. Figure 2b depicts the chemoschedule that produces the closed ABCA cycle, which we use as our adaptive chemoschedule for the finite *N* model, repeating it in a series of cycles. The key to designing a deterministic closed cycle (see [19,26] for more discussion) is to realize that (i) in the absence of chemotherapy, all trajectories converge to the *S* corner; (ii) with $C_{1}$ on (above threshold levels) and $C_{2}$ off, all trajectories converge to the $R_{1}$ corner, and (iii) with $C_{2}$ on (above threshold levels) and $C_{1}$ off, all trajectories converge to the $R_{2}$ corner. These three facts make it straightforward to identify trajectories that cross, and then by turning off and on the drug schedules at precisely the times when they cross (points *A*, *B*, and *C*), one can create a closed cycle. These closed cycles are plentiful and can be designed so that they are located throughout the triangular domain, with a large range in the sizes of the enclosed areas [19]. The results described in this section do not depend, in any crucial way, on which closed cycle we choose in the tri-linear plane and can be viewed as general.

What is the size of the error made in using the stochastic (finite *N*) model with the adaptive chemoschedule that puts us on the closed ABCA cycle in the deterministic problem? In Figure 3a, we show the spread of the endpoints in the $\left(S,R_{1},R_{2}\right)$ tri-linear plane after one evolutionary cycle, for the case e=0 when the two drugs act additively on the population. The cycle is initiated at $\left(S,R_{1},R_{2}\right)=\left(0.8,0.1,0.1\right)$ (point marked *A* in Figure 2), for 10,000 trials, with *N* = 100,000 cells. Additionally shown are the two orthogonal principal components associated with the spread of the data, which we then map, in Figure 3b, to the two principal axes. The kernel density estimates (KDE) for the distributions are shown in Figure 3c,d. The distributions, after one loop, closely resemble multivariate Gaussian distributions in the principal components.

After repeated loops of the cycle, the multivariate Gaussian distribution starts to break down, as shown clearly in the principal component coordinates (Figure 4), which shows the KDE after 2 loops (Figure 4a), 3 loops (Figure 4b), 4 loops (Figure 4c), and 5 loops (Figure 4d). After the third loop, the distribution no longer approximates a multi-Gaussian distribution.

To examine why the multi-Gaussian nature of the distribution breaks down, we show in Figure 5 histogrammed distributions of the three sub-population frequencies. By the third loop, the distributions start to impinge on 0 (one of the sides of the tri-linear plane) or 1 (one of the corners of the tri-linear plane), indicating that a sub-population either vanishes or saturates, distorting the multi-Gaussian spread. By the fourth and fifth loops, the distributions start to converge at these endpoints and the distribution no longer resembles a multi-Gaussian.

The histogrammed distributions for synergistic (e>0) and antagonistic (e<0) drug interactions are shown in Figure 6a,b. In both cases, the distribution begins to distort after roughly three loops, with the synergistic interactions breaking down slightly sooner in the third loop. This is shown nicely in Figure 7 which depicts the two principal components (on a log–linear plot) as a function of cycle number for additive, synergistic, and antagonistic interactions. In all three cases, the variance of the distribution grows exponentially with loop number. Antagonistic drug interactions outperform additive interactions, which both outperform synergistic interactions. One could argue this is because antagonistic interactions are able to suppress competitive release of the resistant populations more efficiently than synergistic interactions, as discussed in [26]. As N→∞, the variances will go to zero.

What effect does this breakdown have on the ability of the model to suppress competitive release and control tumor growth? For this, we use the tumor growth Equation (Equation 16) as shown in Figure 8 where we plot treated vs. untreated volumetric tumor growth using adaptive scheduling. The solid lines depict the deterministic growth curves (tumor recurrence occurs around t≈50), while the error bars show the spread of the adaptive stochastic runs. Our basic message from this plot is that while the stochastic spread (associated with tumor growth) increases as time progresses, it is perhaps not as much as one might expect from the fact that the adaptive schedule is not producing closed evolutionary cycles.

## 4. Discussion

The exponential increase in the distribution variance with each adaptive cycle degrades the precision of each subsequent round of chemotherapy since the cycles were designed to stay in a closed evolutionary loop only in the deterministic limit N→∞. Ideally, the time-dependent chemoschedule keeps the three sub-populations in long-term competition with each other to avoid chemoresistance indefinitely, although delaying chemoresistance through several cycles may be all that is practically achievable due to stochastic fluctuations. When examining the tumor growth curves (Figure 8), we can see that the mismatch between the deterministic chemoschedule, when run in a finite-cell stochastic model, gives results that are nonetheless better than what could be expected given the exponential increase in variance. Nonetheless, we point to several concrete strategies that could help extend the timeframe over which the schedules could remain precise, even if the evolutionary loops do not close. Three strategies are discussed in [36]. The first is to explore other possibilities than the commonly used and accepted 50% rule which identifies a nominal baseline tumor size and advocates applying chemotherapy until the size shrinks by half. Then, when the tumor reaches the nominal baseline size, chemotherapy is again implemented for another cycle, etc. It is easy to imagine that there could be better, more optimized options, but none have been clinically tested as far as we are aware. The second is to explore the role of tumor size in deciding when to apply chemotherapy. Typically, chemotherapy commences as soon as a tumor is discovered, but it is conceivable that this may not be optimal in all cases. In [5], we explored using different chemotherapy schedules for different tumor growth rates, but these ideas have not yet been tested in the clinic. The third idea discussed in [36] is to explore the optimal frequency distribution of the cell populations in which to begin adaptive therapy. From the point of view of our mathematical model, these ideas generally relate to the size and location of the ABCA cycle in the tri-linear phase plane that would be optimal for an adaptive schedule. We, of course, have discretion over both where in the tri-linear plane the cycle starts (location of point *A* in Figure 2a), and how large an area the evolutionary loop ABCA should enclose, both of which are tied to the suggestions made in [36]. Optimizing these decisions could very well have medical implications that could and should be explored further.

## 5. Conclusions

The lessons learned in probing our finite-cell stochastic model using a chemoschedule design from a deterministic model are both sobering and encouraging. On the one hand, it is clear that designing an adaptive schedule using the deterministic model is far more practical than designing a tailored one for each individual stochastic run. The simplicity of this approach, however, comes at the price of not having exactly the right timing and dosing levels for any single realization, which leads to a mismatch between the designed closed loop and the stochastic sub-population levels. This, in turn, leads to an exponential increase in the error variances associated with repeated evolutionary loops meant to keep the sub-populations in perpetual competition. The good news is that the effect that this mismatch has on tumor growth curves seems small—it is possible that the chemotherapy cycles will still be good enough to delay resistance in a wide range of patient populations and tumor sizes. Of course. this is based on the results of a mathematical model and computer simulations, not clinical trials in real-world settings, and should be interpreted in that context.

## Figures and Tables

**Figure 1 cancers-13-02880-f001:**
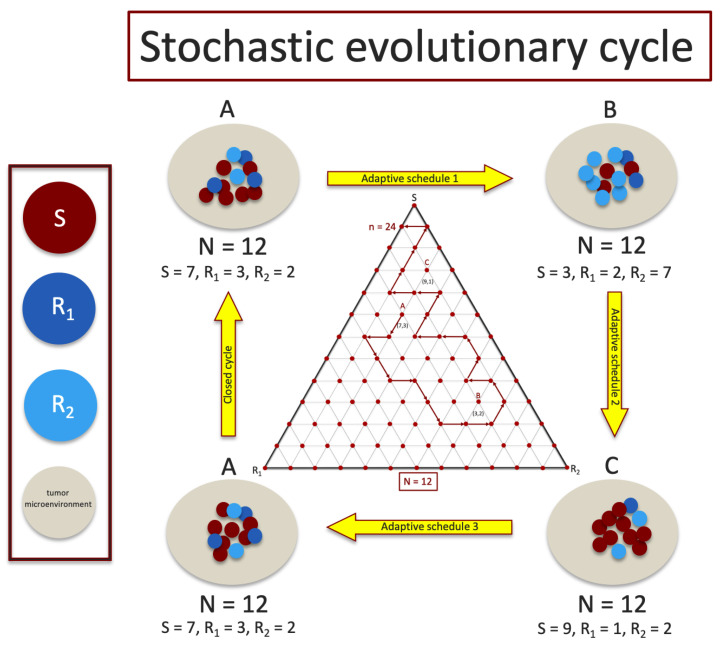
A sequence of adaptive chemoschedules that lock the tumor in a closed evolutionary cycle is difficult to achieve for finite cell (shown for N=12) populations since sub-population frequencies fluctuate stochastically and are difficult to measure with precision. Middle inset shows a discrete tri-linear plot of a stochastic realization in the $\left(S,R_{1},R_{2}\right)$ plane, with n=24 steps, starting at point *A*, using the chemoschedule designed from the deterministic (N→∞) model.

**Figure 2 cancers-13-02880-f002:**
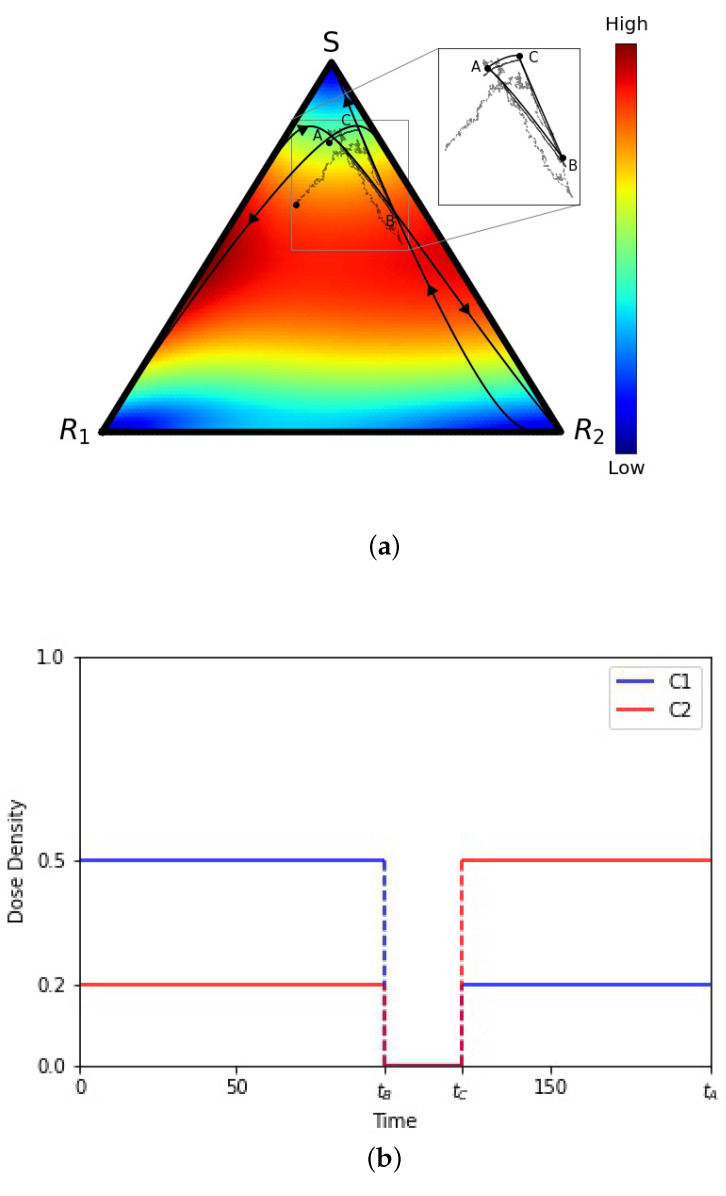
(**a**) Tri-linear $\left(S,R_{1},R_{2}\right)$ coordinate representation of a deterministic evolutionary cycle ABCA, from the adjusted replicator equation, and two stochastic realizations (from the Moran process) with *N* = 10,000 and *N* = 1,000,000 cells. Inset shows that the stochastic paths are not closed. Background colors show velocity field, hence instantaneous speed of convergence, from the adjusted replicator system. (**b**) Adaptive chemotherapy schedule with drugs 1 and 2 for closed cycle ABCA.

**Figure 3 cancers-13-02880-f003:**
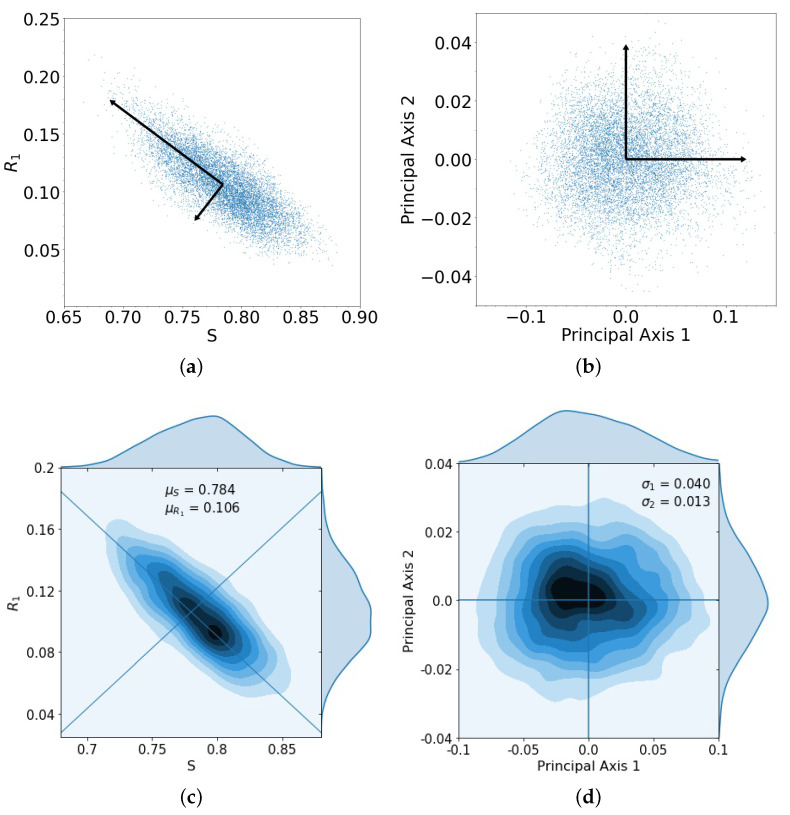
(**a**) Spread of 10,000 trials starting at initial conditions S=0.8, $R_{1}=0.1$, and $R_{2}=0.1$ with principal axis after 1 loop; (**b**) same spread of trials from (**a**) plotted with principal component axes; (**c**) kernel density estimation (KDE) for trials in (**a**), with darker areas indicating a higher concentration of data, with means $\mu_{S}=0.784$, $\mu_{R1}=0.106$; (**d**) the spread of trials closely resembles a multivariate Gaussian distribution composed of the principal components, with singular values $\sigma_{1}=0.040$, $\sigma_{2}=0.013$.

**Figure 4 cancers-13-02880-f004:**
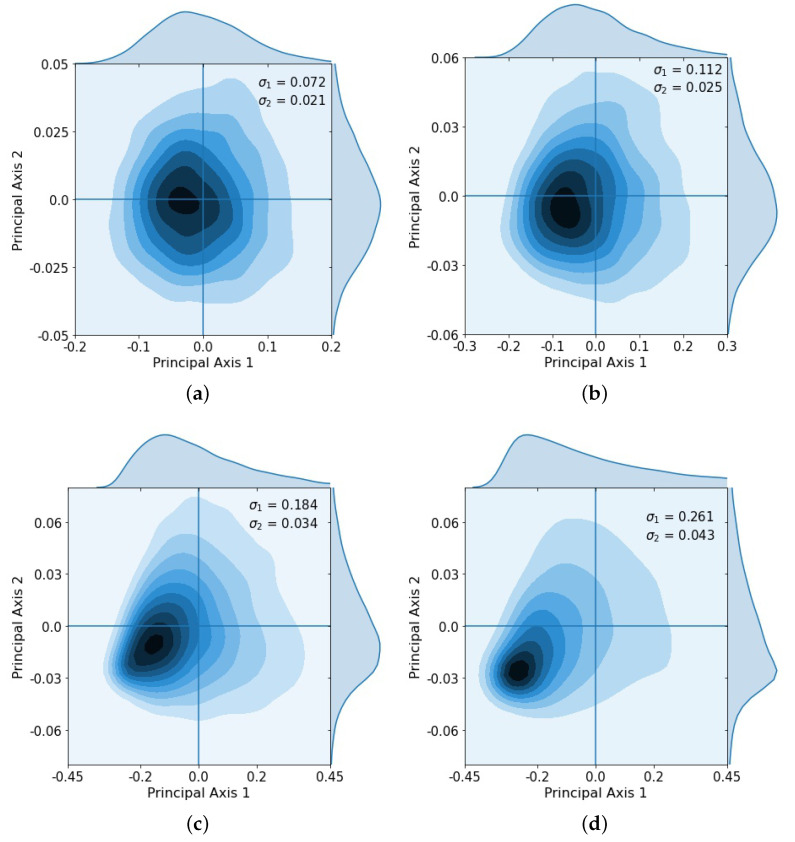
Kernel density estimation showing distribution after successive loops and singular values $\sigma_{1}$, $\sigma_{2}$. Gaussian spread starts to break down after loops 3 and 4 as some tumors began to saturate. (**a**) Loop 2, $\sigma_{1}=0.072$, $\sigma_{2}=0.021$; (**b**) Loop 3, $\sigma_{1}=0.112$, $\sigma_{2}=0.025$; (**c**) Loop 4, $\sigma_{1}=0.184$, $\sigma_{2}=0.034$; (**d**) Loop 5, $\sigma_{1}=0.261$, $\sigma_{2}=0.043$.

**Figure 5 cancers-13-02880-f005:**
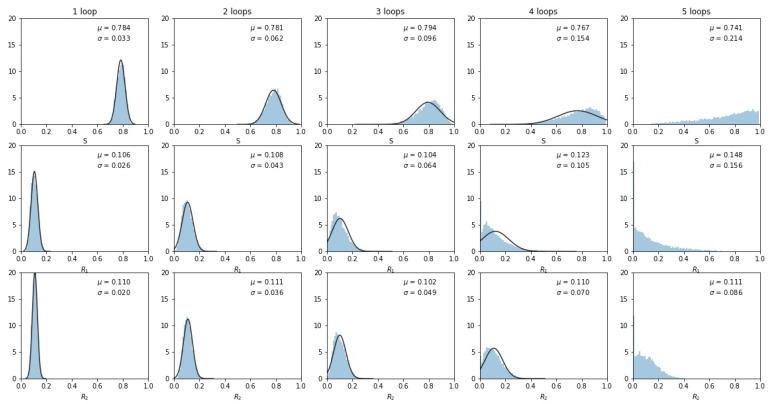
Histogrammed distributions of the three sub-populations as the number of loops increases. Note that some tumors began to fill at the S=1 and $R_{1}=1$ corners, distorting the multivariate Gaussian nature of the distributions. Mean μ and standard deviations σ are shown.

**Figure 6 cancers-13-02880-f006:**
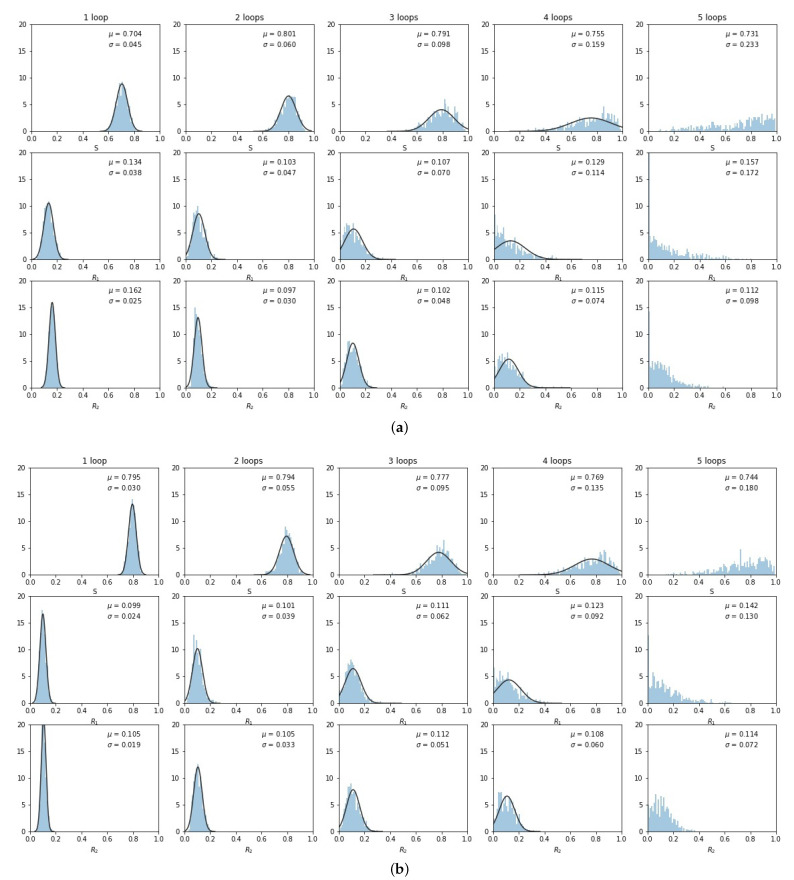
Same as Figure 5, but for synergistic e>0 and antagonistic e<0 drug interactions. (**a**) e=0.3; (**b**) e=−0.3.

**Figure 7 cancers-13-02880-f007:**
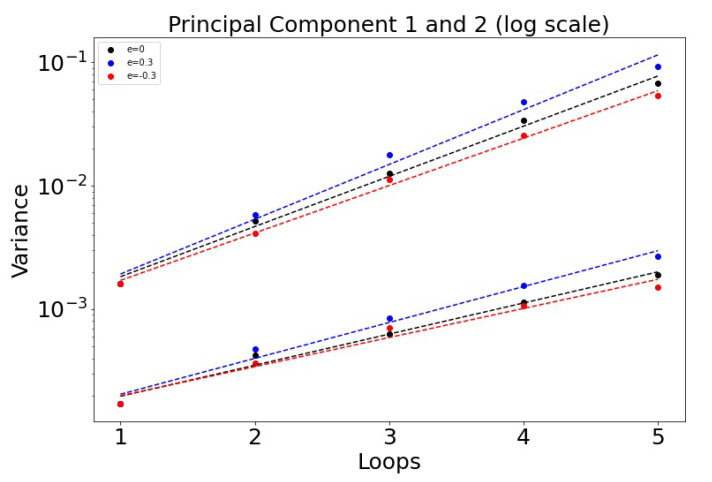
Variance $\sigma_{e}$ of the principal components as a function of the cycle number *n* plotted on log–linear axes, so $\sigma_{e}∼a\exp\left(\alpha_{e}n\right)$. Note that antagonistic interactions grow the slowest, while synergistic interactions grow the fastest. $\alpha_{0}$: 0.936 (PCA 1) and 0.580 (PCA 2). $\alpha_{0.3}$: 1.02 and 0.670. $\alpha_{-0.3}$: 0.885 and 0.543.

**Figure 8 cancers-13-02880-f008:**
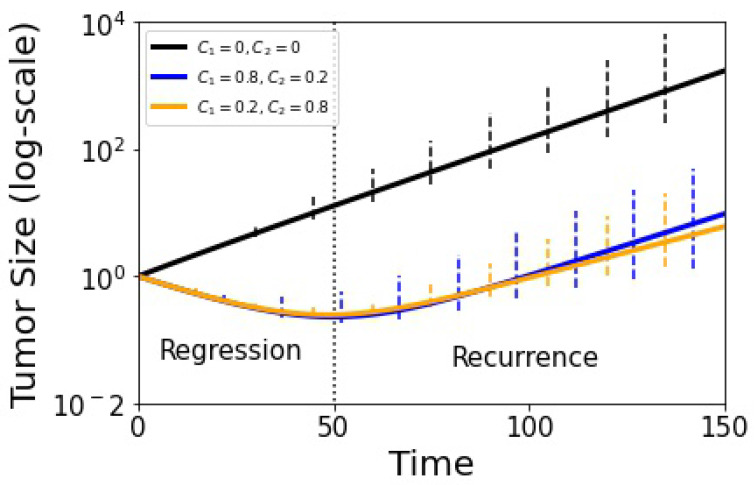
Tumor growth curve with adjusted replicator model using g=1.05 and the stochastic Moran process model (100 runs). The black curve shows untreated growth, blue and yellow show adaptive therapy results. Error bars indicate 90% confidence level. Tumor recurrence sets in at t≈50 in dimensionless time units.

## Data Availability

Available on request to the corresponding author.

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
