# Peer review of "Are Adaptive Chemotherapy Schedules Robust? A Three-Strategy Stochastic Evolutionary Game Theory Model"

_cancers, 2021, doi:10.3390/cancers13122880_

Round 1
Reviewer 1 Report
The manuscript by Dua and colleagues investigates the robustness of two-drug adaptive therapy using a 3-strategy evolutionary game theory model. They design the adaptive therapy so that it results in closed evolutionary loops in the deterministic (replicator equation) limit, and they then apply the therapy repeatedly in a finite-N stochastic version of the model. They find that the resulting error distributions are approximately Gaussian for the first few cycles, but that breaks down at later cycles as resistant subpopulations saturate the tumor. The work suggests that adaptive therapies (particularly those based on synergistic drug pairs) may be difficult to design in practice, as variance increases exponentially over time.
The work addresses an important problem and is technically sound. It draws on the established connection between stochastic (finite-N) and deterministic replicator models, using the latter to identify cycles and the former (formulated as a discrete time Markov chain) to study time-dependent variance. Overall, the study will be of interest to those studying adaptive therapies and drug interactions, both in cancer and beyond (e.g. antibiotic resistance).
I have the following (minor) comments that might improve the manuscript:
As I understand it, the specific dosing schedule is chosen so that the system forms a closed evolutionary loop in the deterministic limit. However, I don’t believe the authors discuss how they identify these closed loops in the first place. Can the authors comment briefly on the nature of these cycles? For example, do such control cycles always exist in this Prisoner’s dilemma version of the replicator model? How many such cycles are there? Any comments on what is known about the controllability of this system would help guide the uninitiated reader (I believe some of these are discussed in previous work by the same authors, and it might help to briefly discuss here).
Related to the first question: I believe the results presented are all based on one particular cycle in phase space. To what extent do the authors believe the results are generalizable to other cycles? And how might the qualitative results change as the model parameters change (e.g. changing the entries of the payoff matrix but retaining the necessary inequalities for the PD).
In some sense, the results presented might be overly pessimistic, given that the cycles are not adjusted in real time to account for the current state of the system (instead, the cycle is determined up front from the deterministic dynamics, if I understand correctly). While perhaps technically impractical, in principle couldn’t the adaptive therapies be improved, even in the finite-N case, by using a type of stochastic control (like in ref 7, or using something like a markov decision process: https://journals.plos.org/plosbiology/article?id=10.1371/journal.pbio.3000515). In other words, the breakdown here is not necessarily saying that the system can’t be controlled with adaptive therapy, but instead says that the stochasticity of the trajectory leads to a mismatch between the true and predicted state of the system, leading to a mismatch between the applied control (drug) and the optimal control. It might improve the paper to discuss this point in the discussion.
I would add a sentence or two discussing in a bit more detail why the Prisoner’s dilemma is the appropriate game to consider. These points are discussed in detail in the references, but I think it would help the general reader to add a sentence or two more when the PD is first introduced.
Since the system becomes deterministic in the large N limit, I would guess that the breakdown of the adaptive therapy depends sensitively on N (the exponential increase in variance may persist even for very large N). Can the authors briefly comment on their choice of N and how results might be expected to change as N varies?
Author Response
As I understand it, the specific dosing schedule is chosen so that the system forms a closed evolutionary loop in the deterministic limit. However, I don’t believe the authors discuss how they identify these closed loops in the first place. Can the authors comment briefly on the nature of these cycles? For example, do such control cycles always exist in this Prisoner’s dilemma version of the replicator model? How many such cycles are there? Any comments on what is known about the controllability of this system would help guide the uninitiated reader (I believe some of these are discussed in previous work by the same authors, and it might help to briefly discuss here).
Response: We do this now in the revised Results section 3, and also point to separate publications[19],[26] where our method of finding closed loops and their controllability is discussed further.
Related to the first question: I believe the results presented are all based on one particular cycle in phasespace. To what extent do the authors believe the results are generalizable to other cycles? And how might the qualitative results change as the model parameters change (e.g. changing the entries of the payoff matrix but retaining the necessary inequalities for the PD).
Response: We comment on this now in the results section – our results generally do not depend on which closed cycle we choose in any special or delicate way. As far as the PD entries, yes, we need to use that matrix for relevance to cancer modeling, but as far as designing closed loops, we have not explored other payoff matrices at this point.
In some sense, the results presented might be overly pessimistic, given that the cycles are not adjusted in real time to account for the current state of the system (instead, the cycle is determined up front from the deterministic dynamics, if I understand correctly). While perhaps technically impractical, in principle couldn’t the adaptive therapies be improved, even in the finite-N case, by using a type of stochastic control (like in ref 7, or using something like a Markov decision process:
https://journals.plos.org/plosbiology/article?id=10.1371/journal.pbio.3000515). In other words, the breakdown here is not necessarily saying that the system can’t be controlled with adaptive therapy, but instead says that the stochasticity of the trajectory leads to a mismatch between the true and predicted state of the system, leading to a mismatch between the applied control (drug) and the optimal control. It might improve the paper to discuss this point in the discussion.
Response: Very good point. We now discuss this with regards to the tumor growth equation and it is true that even though the evolutionary loops are not exactly closed, the tumor growth equation seems to still show huge improvements using the adaptive schedules.
I would add a sentence or two discussing in a bit more detail why the Prisoner’s dilemma is the appropriate game to consider. These points are discussed in detail in the references, but I think it would help the general reader to add a sentence or two more when the PD is first introduced.
Response: We now do this and refer to an additional publication as well.
Since the system becomes deterministic in the large N limit, I would guess that the breakdown of the adaptive therapy depends sensitively on N (the exponential increase in variance may persist even for very large N). Can the authors briefly comment on their choice of N and how results might be expected to change as N varies?
Response: I would not describe the breakdown as depending sensitively on N. I would describe it as depending very predictably on N, going to 0 as N goes to infinity (most likely like 1/\sqrt(N), but we have not pursued this as it is not particularly relevant to the goals of this manuscript). But yes, we now comment on this point.
Reviewer 2 Report
In this paper, authors develop a three-component Moran process, which assumes the total number of cells is constant, to quantify the variance of the response to treatments. The cell population consists of chemosensitive cells, chemoresistant cells to drug 1, and chemoresistant cells to drug 2, and the drug interactions can be synergistic, additive, or antagonistic.
As a result, they found that as the number of cycles increases, one of the three subpopulations typically saturates the tumor (competitive release) resulting in treatment failure in the finite cell population model. They suggest that designing an effective and repeatable adaptive chemoschedule in practice will require a highly accurate tumor model and accurate measurements of the subpopulation frequencies or the errors will quickly (exponentially) degrade its effectiveness, particularly when the drug interactions are synergistic.
I found the paper very poorly written, no appropriate introduction and lack of a good discussion. The methods and results sections are worse. The model is hard to follow as the variables of the model are not defined, and the equations are given without description. More importantly, the main assumption of the model, which is the number of cells stays constant during treatment, is not valid for cancer treatments. I therefore cannot recommend this article for publication.
Major points:
Moran model is not appropriate for modeling cancer treatments, because the main assumption of this model, which is the constant number of cells, does not hold in any cancer treatment especially chemotherapy. Cells die and they are not in any of these three components that have been modeled.
In section 2, variables of the model, e.g. R1, R2, w1, w2, and S should be defined before the formula is given.
There is no description about formulas. For example, why should the relative fitness of the three subpopulations be controlled by the selection pressure parameters?
Results section has been written poorly. It just says what each figure shows; like series of sentences about the caption of figures without properly analyzing them.
Author Response
I found the paper very poorly written, no appropriate introduction and lack of a good discussion. The methods and results sections are worse. The model is hard to follow as the variables of the model are notdefined, and the equations are given without description. More importantly, the main assumption of the
model, which is the number of cells stays constant during treatment, is not valid for cancer treatments. I therefore cannot recommend this article for publication.
Response: Thank you for this criticism which is well taken. Aside from polishing some of the writing, we have added a Model description section 2, re-written the Results section 3, and added a Conclusion section 5. We believe this version is greatly improved because of these changes which address all of your
points.
Moran model is not appropriate for modeling cancer treatments, because the main assumption of this model, which is the constant number of cells, does not hold in any cancer treatment especially chemotherapy. Cells die and they are not in any of these three components that have been modeled.
Response: We did not make clear in the earlier version of the manuscript that in addition to the fixed N Moran process which tracks frequencies, we also have a tumor volumetric growth equation (16) in which tumor volume grows or decays based on the average fitness of the tumor cell population. This additional
equation coupled to the Moran process is what completes the tumor description. You are correct, without it, the model would only redistribute frequencies with N fixed.
In section 2, variables of the model, e.g. R1, R2, w1, w2, and S should be defined before the formula is given.
Response: We do this now in the Model description section.
There is no description about formulas. For example, why should the relative fitness of the three subpopulations be controlled by the selection pressure parameters?
Response: We now explain this more thoroughly.
Results section has been written poorly. It just says what each figure shows; like series of sentences about the caption of figures without properly analyzing them.
Response: The results section has now been completely re-written.
Reviewer 3 Report
Using game theory, the authors evaluated the effectiveness of adaptive chemotherapy in overcoming chemoresistance..
Optimal adaptive treatment cycles were test on a virtual patient population for their robustness.
The study highlights the significance of accurate measurements of tumor cell populations and an appropriate tumor modeling framework to design effective adaptive chemotherapy schedules.
The paper is well written and novel enough to warrant publication.
Author Response
Using game theory, the authors evaluated the effectiveness of adaptive chemotherapy in overcoming chemoresistance. Optimal adaptive treatment cycles were test on a virtual patient population for their robustness. The study highlights the significance of accurate measurements of tumor cell populations and an appropriate tumor modeling framework to design effective adaptive chemotherapy schedules.The paper is well written and novel enough to warrant publication.
Response: See changes made as detailed above based on referee comments from others.
Round 2
Reviewer 2 Report
The manuscript has been remarkably improved.